# Food Security and Climate Stabilization: Can Cereal Production Systems Address Both?

**Long Liang [1,*], Bradley G. Ridoutt [2,3] and Liyuan Wang [4]**

1   The Strategy Research Institute of Rural Revitalization, Guizhou University of Finance and Economics, Guiyang 550025, China
2   Commonwealth Scientific and Industrial Research Organisation (CSIRO) Agriculture and Food, Clayton South, VIC 3168, Australia; brad.ridoutt@csiro.au
3   Department of Agricultural Economics, University of the Free State, Bloemfontein 9300, South Africa
4   Shanghai Academy of Agricultural Science, Shanghai 201403, China; liyuanw007@163.com
*   Correspondence: txws0109@126.com

**Abstract:** There is abundant evidence that greenhouse gas (GHG) emissions of cereal products, expressed per ton of grain output, have been trending downward over the past 20 years. This has largely been achieved through agricultural intensification that has concurrently increased area-based GHG emissions. The challenge is for agriculture to increase grain yields to meet the food demands of a growing world population while also contributing to climate stabilization goals by reducing net GHG emissions. This study assessed yield-based and area-based emissions and efficiencies for the winter wheat–summer maize (WWSM) rotation system over the period 1996 to 2016 using long-term, longitudinal, farm survey data and detailed soil emission data in Huantai county, Shandong Province, which is an archetype for cereal production across the North China Plain (NCP). In this region, yields have been increasing over time. However, nitrogen fertilizer inputs have decreased substantially with greater adoption of soil nutrient testing. In addition, there has been widespread adoption of residue incorporation into soils. As such, since 2002, the product carbon footprints of wheat and maize have reduced by 25% and 30%, respectively. Meanwhile, area-based carbon footprints for the rotation system have reduced by around 15% over the same period. These findings demonstrate the importance of detailed assessment of soil $N_2O$ emissions and rates of soil organic carbon sequestration. They also show the potential for net reductions in GHG emissions in cropping without loss of grain yields.

**Keywords:** agricultural soils; GHG emission; life cycle assessment; product carbon footprint; carbon efficiency; agricultural intensification; fertilizer management

## 1. Introduction

Agriculture is simultaneously facing the challenges of increasing yields while also reducing environmental impacts [1–5]. In this regard, the management of fertilizer inputs is important as high yielding crops depend on adequate nutrition, however there are considerable environmental costs associated with fertilizer production and use, such as greenhouse gas (GHG) emissions [2,3,6]. Around 25% of global GHG emissions are attributed to land use change, crop production, and fertilizer manufacture and use [2,7]. In China, it has been identified that there is potential for a 30% to 50% increase in grain yields without increasing fertilizer inputs, if cropping systems are improved [4,8–10]. In addition, well-managed agricultural soils have GHG sequestration capability [11]. Lal [12] suggested that carbon sequestration in agricultural systems has the potential to offset between 5% and 15% of global fossil-fuel emissions. As such, agriculture has a strategic role to play in GHG emissions management as well as food security.

Environmental indicators such as the carbon footprint (CF) and carbon efficiency (CE) are often used to evaluate the sustainability of agricultural production systems [5,13,14].

The CF evaluates the balance of GHG emissions and sequestrations from a product or system across its life cycle [15]. In the case of crop production, this includes the GHG emissions associated with the production of farming inputs, such as fertilizer. A variety of CE metrics have been proposed, typically expressing yield or value created relative to emissions [16].

Wheat is an important crop globally, and CFs have now been reported for production in many regions [16–20]. Meisterling et al. [21] and Knudsen et al. [22] compared the CF of conventional and organic wheat production. Röös et al. [23], Espinoza-Orias et al. [24], and Meul et al. [25] identified the CFs of products derived from wheat, including pasta, bread, and animal feed. Other studies have evaluated alternative farming practices [26–28]. Global estimates and comparisons between countries have also been undertaken [29–31]. In China, there have also been many assessments of the CF of crop production systems [5,32–38]. As for carbon efficiency, Lal [16], Maheswarappa et al. [39], and Aweke et al. [40] used this indicator to evaluate the sustainability of agricultural ecosystems in USA, India, and Ethiopia, respectively. Taking Punjab and Ohio as examples, Dubey and Lal [14] made a comparison of the agricultural production systems of India and America. In China, Shi et al. [41], Long [42], Cheng et al. [13], Tian et al. [43], and Yin et al. [44] used CE to evaluate the production efficiency of farmland.

The case study evidence based on CF and CE is large. Direct comparisons between studies are not straightforward due to different modeling choices [26,29,34,44–49]. Nevertheless, taken together, the evidence suggests that over the past few decades the yield-scaled carbon emissions of cereal production have been reducing while the area-scaled carbon emissions are still increasing [50–52], and this trajectory is likely to continue into the future. Reductions in GHG emissions are valuable. However, this does not address GHG emission in aggregate, which need to also be reduced if climate stabilization is to be achieved.

Continued intensification of farming systems is unlikely to address this problem. New farming system models are needed that enable high grain yields to be achieved while also achieving a reduction in net GHG emissions per unit of cropland area. In this study, taking Huantai county, north China, as an example of a classic high-yielding crop production region, we make a detailed CF and CE analysis based on experimental and survey data over the period 1996 to 2016. The aim of this study is to analyze the status and trend of cereal production and to identify pathways to increasing yield while also reducing area-based emissions. We seek to contribute insights relevant to the development of sustainable farming systems that can contribute to both food security and climate stabilization goals.

## 2. Material and Method

### 2.1. Study Area

Huantai county (36°51′50″–37°06′00″ N, 117°50′00″–118°10′40″ E, Figure 1) is located in the center of the Shandong Province, which is a part of the North China Plain (NCP). This region covers an area of 509 km$^2$ and includes around 0.5 million people, of which 0.43 million live in rural communities. It is a typical continental monsoonal climate, with a mean altitude of 6.5–29.5 m, and the average annual temperature and precipitation are 12.5 °C and 580 mm, respectively. The main soil types include Hapludalfs, Aquents, and Vertisols [53,54]. This region is within the primary cereal-producing area of China, and more than 80% of agricultural land use between 1980 and 2016 has adopted a winter wheat (*Triticum aestivum* L.)–summer maize (*Zea mays* L.) (WWSM) rotation system. The yield in 1990 was >15 Mg/ha of grains across the entire region. Thus, this county became the first grain county in northern China, and cereal production has been intensified in this region since 1990. To some extent, Huantai county is representative of the larger NCP.

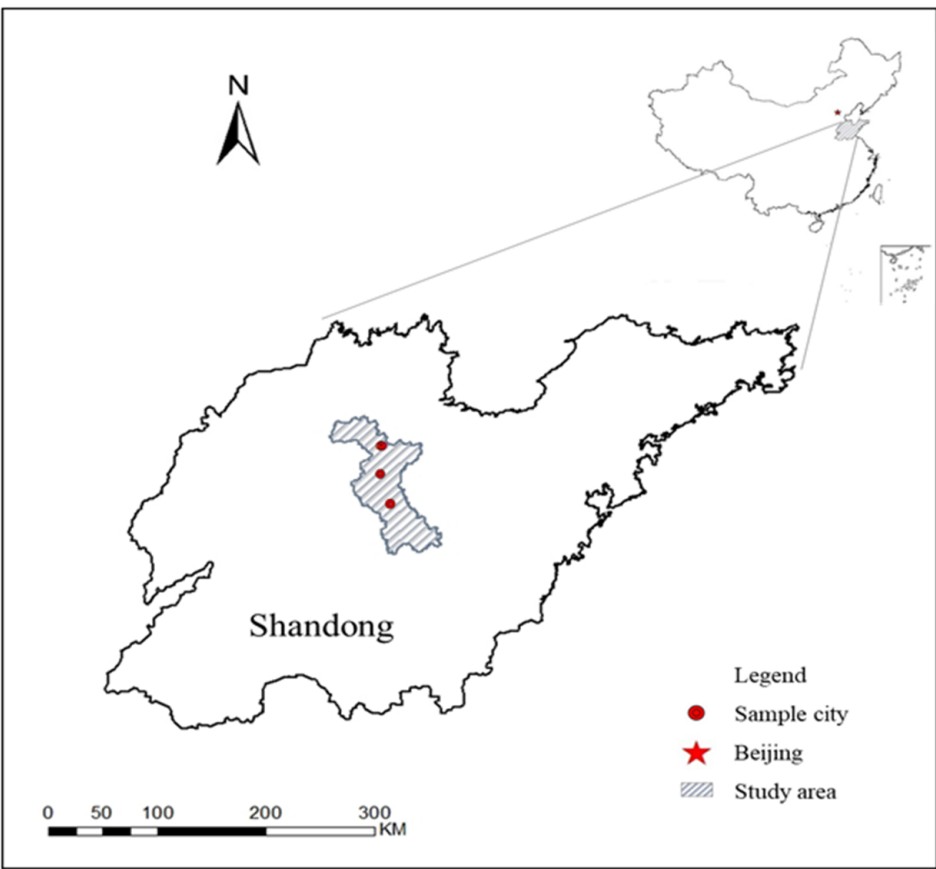

**Figure 1.** The study area of Huantai county, Shandong Province in China.

*2.2. Data Collection*

The data used in this study came from local agricultural surveys and experiments. In 1997, 2003, 2007, 2013, and 2017, teams from China Agricultural University undertook studies of agricultural production in Huantai county. These investigations were carried out in the same three towns (Tanshan, Chengzhuang, and Guoli), using an identical questionnaire. For each town, two villages were randomly chosen and 20 farming households were investigated from each village as a stratified random sample. As a prerequisite, towns and villages were only selected if most of the farmlands employed the WWSM rotation system. Selections were also made to achieve a cross section of higher and lower levels of productivity. With the passage of time, around 50% of agricultural lands in the region have been gradually consolidating, and the scale per farm was 6.23 ha, with fewer peasant farmers. As such, in 2013 and 2017, the goal of surveying 20 households was replaced with a goal of surveying households responsible for at least 80% of cropping in each village. Table 1 presents a summary of farming inputs and outputs over the period 1996 to 2016. In addition, at the county level, data concerning fertilizer inputs and grain outputs were obtained from the Huantai statistical yearbooks compiled by the local government.

In 2007, China Agricultural University and Huantai county also jointly constructed an ecological and sustainable development experiment station. A series of experiments were deployed in the station, and the longest experiment has exceeded 10 years [55]. Data related to soil emissions and carbon sequestration were obtained from these long-term experiments [11,54,56,57].

**Table 1.** Inputs and outputs per hectare of the winter wheat–summer maize (WWSM) rotation system in Huantai county, Shandong Province, China from 1996 to 2016.

| Item | 1996 | | 2002 | | 2006 | | 2012 | | 2016 | |
|---|---|---|---|---|---|---|---|---|---|---|
| | Wheat | Maize | Wheat | Maize | Wheat | Maize | Wheat | Maize | Wheat | Maize |
| Seed (kg) | 166.1 | 50.1 | 140.8 | 46.2 | 111.5 | 40 | 121 | 24.2 | 121 | 27.8 |
| N (kg) | 322.1 | 286.9 | 349.2 | 369.8 | 260.7 | 292.7 | 214.7 | 254.3 | 222 | 231.9 |
| $P_2O_5$ (kg) | 131.2 | 37.5 | 264.1 | 173.2 | 179.6 | 164.5 | 167.7 | 132.2 | 108 | 86.3 |
| $K_2O$ (kg) | 33.9 | 30.2 | 29.8 | 49 | 76.9 | 130.9 | 46.8 | 53.7 | 70.5 | 71.3 |
| Irrigation ($m^3$) | 3821 | 1597 | 3352 | 2362 | 3960 | 2640 | 3257 | 2059 | 3375 | 1875 |
| Electricity (kWh) | 1146 | 479 | 1006 | 709 | 1188 | 792 | 977 | 618 | 972 | 596 |
| Diesel (kg) | 128.1 | 131.8 | 137.6 | 204.2 | 157.9 | 274 | 167.4 | 285.1 | 175.5 | 304.5 |
| Herbicide (kg) | 0.4 | 1.0 | 0.5 | 1.3 | 0.6 | 1.5 | 1.0 | 1.9 | 0.3 | 1.0 |
| Pesticide (kg) | 0.8 | 0.5 | 1.1 | 0.9 | 1.2 | 1.2 | 5.1 | 3.8 | 0.9 | 0.8 |
| Grain (kg) | 6510 | 7630 | 6850 | 7840 | 7052 | 8086 | 7330 | 7520 | 7425 | 9767 |

## 2.3. Functional Unit and System Boundary

For this study of product carbon footprints (PCF) and area carbon footprints (ACF), the units of analysis were one kg of grain and one ha of cropland, respectively. A life cycle assessment approach was adopted [58,59], with the system boundary from cradle to farm gate in order to conveniently compare with similar studies. The considered emissions were $CO_2$, $CH_4$, and $N_2O$. Results were expressed as $CO_2$ equivalent emissions ($CO_2$eq), using the GWP100 (Global Warming Potential aggregated over 100 years) climate metric. The GWP values for $CO_2$, $CH_4$, and $N_2O$ were 1, 25, and 298, respectively, based on The Intergovernmental Panel on Climate Change (IPCC) [60]. According to the suggestion of Adewale et al. [45], the boundary of agricultural CFs considered the following factors: land-use change, machinery and electricity use, fuel consumption, pesticides and other chemical inputs consumption, material inputs, fertilization, soil GHG emission, and soil organic carbon (SOC) sequestration, etc. All these factors can be divided into two parts, namely pre-farm and on-farm subsystems [17]. In this study, the former include the production and transportation of electricity, fuel, fertilizers, chemicals, machinery, and irrigational facility, and the latter include machinery operation, soil emission, SOC sequestration, etc.

## 2.4. Carbon Footprint Calculation Method

### 2.4.1. Pre-Farm Subsystem

GHG emission factors were chosen that most accurately reflect the local production systems and sources of farming inputs used (Table 2). For the pre-farm subsystem, CFs were calculated according to Equation (1).

$$CF_{input} = \sum Q_i \times EF_i \qquad (1)$$

where, $CF_{input}$ is the total amount of carbon footprint due to the production, transportation, and application of agricultural inputs (kg $CO_2$eq/ha/season), $Q_i$ is the quantity of an $i$th individual agricultural input used in wheat and maize production season (kg/ha/season), and $EF_i$ is the emission factor of each input (kg $CO_2$eq/kg).

**Table 2.** Life cycle greenhouse gas (GHG) emission factors.

| Item | Emission Factor | Unit | References |
|---|---|---|---|
| $CH_4$ | 25 | kg $CO_2$eq/kg | [60] |
| $N_2O$ | 298 | kg $CO_2$eq/kg | [60] |
| Seed | 1.18 | kg $CO_2$eq/kg | [58] |
| N | 8.3 | kg $CO_2$eq/kg N | [32] |
| $P_2O_5$ | 1.5 | kg $CO_2$eq/kg $P_2O_5$ | [32] |
| $K_2O$ | 0.98 | kg $CO_2$eq/kg $K_2O$ | [32] |
| Electricity | 0.92 | kg $CO_2$eq/kWh | [58] |
| Diesel | 3.32 | kg $CO_2$eq/kg | [58] |
| Irrigation facilities | 220 (110) | kg $CO_2$eq/ha | [42] |
| Machine | 6.74 | kg $CO_2$eq/kg | [58] |
| Herbicide | 18 | kg $CO_2$eq/kg | [58] |
| Pesticide | 18 | kg $CO_2$eq/kg | [58] |

Note: The value in parenthesis refers to the maize production season.

2.4.2. On-Farm Subsystems

As for on-farm subsystems, the *CF* of diesel consumption by machine operation was calculated by Equation (1). Soil $N_2O$ emissions were calculated according to Zhang et al. [57] using Equations (2) and (3):

$$\text{Wheat}: CF_{N2O} = \left(0.0052 \times N_{input} + 0.6435\right) \times 298 \tag{2}$$

$$\text{Maize}: CF_{N2O} = \left(0.0101 \times N_{input} + 0.6003\right) \times 298 \tag{3}$$

where, $CF_{N2O}$ is the cumulative amounts of $N_2O$ emission by the soil caused by *N* fertilizer application in the wheat and maize production seasons (kg $CO_2$eq/ha/season); $N_{input}$ is the *N* fertilizer application of wheat and maize production; 298 is the coefficient for converting $N_2O$ to $CO_2$eq; and 0.0052, 0.0101, 0.6435, 0.6003 are the related emission coefficients.

Zhao et al. [11] and Zhao [56] quantified $CH_4$ absorption by agricultural soils in Huantai county with the mean amount at 1.5 kg C/ha/yr. Thus, the *CF* was calculated according to Equation (4).

$$CF_{CH4} = \frac{1}{2}\left(1.5 \times \frac{16}{12} \times 25\right) \tag{4}$$

where, $CF_{CH4}$ is the cumulative amount of absorbed $CH_4$ (kg $CO_2$eq/ha/season), $\frac{16}{12}$ and 25 are the coefficients for converting $C$ to $CH_4$ and $CH_4$ to $CO_2$eq respectively.

Liao et al. [54] demonstrated that agricultural intensification in Huantai county resulted in soil organic carbon (SOC) sequestration based on studies from 1980 to 2011. Thus, the sequestration of annual soil C was calculated according to Equations (5) and (6).

$$SCS = \text{SOC} \times \text{BD} \times \text{H} \times 10 \tag{5}$$

$$\Delta SCS = \frac{1}{2}\left(\frac{SCS_{2011} - SCS_{1980}}{30} \times \frac{44}{12}\right) \tag{6}$$

where, *SCS* is the soil organic carbon sequestration (ton/ha), SOC is the soil organic carbon concentration (7.8 g/kg in 1980 and 11 g/kg in 2011; [54]), BD is the soil bulk density (1.4 g cm$^3$ in 1980 and 1.5 g cm$^3$ in 2011; [54]), H is the thickness of the soil layer (m), and 10 is the coefficient for converting kg/m$^2$ into ton/ha. $\Delta SCS$ is the annual change in SOC storage in a 0–20 cm profile from 1980 to 2011 (kg $CO_2$eq/ha/season), $SCS_{1980}$ and $SCS_{2011}$ are the SOC storage values of the 0–20 cm profile in 1980 and 2011, respectively; 30 is the number of years of the survey period; and $\frac{44}{12}$ is the coefficient for converting C into $CO_2$.

The soil $CO_2$ net flux is estimated to contribute <1% to the global warming potential (GWP) of agriculture on a global scale, which was not considered in this study [3,58].

The area carbon footprint (ACF) and product carbon footprint (PCF) of WWSM rotation system were calculated using Equations (7) and (8).

$$\text{ACF} = CF_{input} + CF_{N2O} + CF_{CH4} + \Delta SCS \tag{7}$$

$$\text{PCF} = \frac{CF_{input} + CF_{N2O+}CF_{CH4} + \Delta SCS}{Y} \tag{8}$$

where, ACF and PCF are the net carbon footprint of wheat and maize production per unit hectare (kg $CO_2$eq/ha/season) and grain production (kg $CO_2$eq/season/kg of grain), *Y* is the grain yield of winter wheat or summer maize (kg/ha/season).

2.5. Carbon Efficiency Calculation Method

Product efficiency (*Ep*), ecological efficiency (*Ec*), and economic efficiency (*Ee*) were calculated using Equations (9)–(12), based on the methods reported by Lal [16] and Shi et al. [41]:

$$Ep = \frac{\text{ACF}}{Y} \tag{9}$$

$$\text{Wheat}: Ec = \frac{\{[Y \times (1 + 1.1)] \times 1.15)\} \times 0.45 \times \frac{44}{12}}{\text{ACF}} \tag{10}$$

$$\text{Maize}: Ec = \frac{\{[Y \times (1 + 1.2)] \times 1.15)\} \times 0.45 \times \frac{44}{12}}{\text{ACF}} \tag{11}$$

$$Ee = \frac{Y \times P}{\text{ACF}} \tag{12}$$

where, $Ep$ is the production efficiency per unit carbon input (kg grain/kg $CO_2$eq), with higher values indicating higher efficiency; $Ec$ and $Ee$ refer to ecological efficiency and economic efficiency, namely the ratio of carbon output (including carbon absorbed by grain, straw, and root) to input (kg $CO_2$/kg $CO_2$eq), with a value >1 indicating the output is higher than the input; and the ratio of economic output to carbon input (Yuan/kg $CO_2$eq). P is the sale price of wheat or maize grain in different years. Furthermore, 1.1 and 1.2 are the ratios of straw to grain for wheat and maize production; 1.15 is the ratio of the total biomass (involving grain, straw, and root) to the shoot (including grain and straw), and 0.45 and $\frac{44}{12}$ are the coefficient of C in biomass and the coefficient for converting C into $CO_2$, respectively.

## 3. Results

### 3.1. Input–Output of Cereal Product System and GHG Emissions

Over the last two decades (1996–2016), N fertilization, electricity use, diesel use, and machinery production have made the largest contributions to the GHG emissions associated with the WWSM cropping system practiced in Huantai county, amounting to 85–88% and 91–94% of the total emissions of wheat and maize production, respectively (Table 3). The largest contribution was from N fertilizer. However, its proportional contribution has been decreasing (Figure 2). For wheat production, N fertilizer CF increased from 2674 kg $CO_2$eq/ha in 1996 to 2898 kg $CO_2$eq/ha in 2002, and then gradually decreased and, in 2016, the N CF was 1843 kg $CO_2$eq/ha. The proportion of GHG emissions related to N inputs decreased from 48% in 2002 to 36% in 2016. Machinery (including manufacture, transportation, use, and maintenance) was the second factor, and its proportion increased from 15% in 1996 to 23% in 2016. Electricity use for irrigation was the third factor, and its contribution ranged from 15% to 20% over the past 20 years. The contribution from diesel fuel consumption increased over time, from 7.5% in 1996 to 11.4% in 2016 (Table 3 and Figure 3).

For maize production, the proportion of GHG emissions related to N inputs also decreased over time from 54% in 1996 to 33% in 2016. In contrast, the contribution from machinery and diesel consumption increased over time, from 19.8% to 34.4% in the case of machinery and from 9.9% to 17.2% in the case of diesel fuel. The contribution from electricity was relatively steady. It is evident that maize production is more dependent on machinery and diesel fuel compared to wheat production (Table 3 and Figure 3).

In the on-farm subsystem, soil is both a GHG source and sink due to N fertilizer input and SOC accumulation (Table 3). The value of $CH_4$ absorbed by the soil was only around 25 kg $CO_2$eq/ha/season and is a relatively less important process. However, $N_2O$ emissions play an important role in the GHG balance of cropping systems. In the process of wheat production, the quantity of soil emissions ranged from 525 to 733 kg $CO_2$eq/ha, and that of soil sequestration was 734 kg $CO_2$eq/ha (Table 3), resulting in net sequestration. In maize production, soil emissions ranged from 877 to 1292 kg $CO_2$eq/ha, exceeding soil sequestration. However, with maize production, the gap between soil emissions and soil sequestration became smaller over time, from 558 kg $CO_2$eq/ha in 2002 to 143 kg $CO_2$eq/ha in 2016. Based on this trajectory, it is possible that soil sequestrations could offset soil emissions in the future.

**Table 3.** Carbon footprint (CF) per hectare (kg $CO_2$eq/ha) and per kilogram grain (kg $CO_2$eq/kg) of WWSM rotation system of Huantai county, Shandong Province, China from 1996 to 2016.

| Item | 1996 | | 2002 | | 2006 | | 2012 | | 2016 | |
|---|---|---|---|---|---|---|---|---|---|---|
| | **Wheat** | **Maize** | **Wheat** | **Maize** | **Wheat** | **Maize** | **Wheat** | **Maize** | **Wheat** | **Maize** |
| Seed | 196 | 59 | 166 | 55 | 132 | 47 | 143 | 29 | 143 | 33 |
| N | 2673 | 2381 | 2898 | 3069 | 2164 | 2429 | 1782 | 2111 | 1843 | 1925 |
| $P_2O_5$ | 197 | 56 | 396 | 260 | 269 | 247 | 252 | 198 | 162 | 130 |
| $K_2O$ | 33 | 30 | 29 | 48 | 75 | 128 | 46 | 53 | 69 | 70 |
| Electricity | 1055 | 441 | 925 | 652 | 1093 | 729 | 899 | 568 | 894 | 549 |
| Diesel | 425 | 438 | 457 | 678 | 524 | 910 | 556 | 947 | 583 | 1011 |
| Pesticide | 21 | 28 | 28 | 40 | 33 | 49 | 108 | 103 | 22 | 32 |
| Irrigation facility | 220 | 110 | 220 | 110 | 220 | 110 | 220 | 110 | 220 | 110 |
| Machine | 851 | 875 | 914 | 1356 | 1049 | 1819 | 1112 | 1893 | 1165 | 2022 |
| Soil emission | 691 | 1042 | 733 | 1292 | 596 | 1060 | 525 | 944 | 536 | 877 |
| $CH_4$ sequestration | −25 | −25 | −25 | −25 | −25 | −25 | −25 | −25 | −25 | −25 |
| Soil sequestration | −734 | −734 | −734 | −734 | −734 | −734 | −734 | −734 | −734 | −734 |
| Grain output (kg) | 6510 | 7630 | 6850 | 7840 | 7052 | 8086 | 7330 | 7520 | 7425 | 9767 |
| ACF −Soil | 5671 | 4418 | 6034 | 6268 | 5559 | 6469 | 5117 | 6011 | 5100 | 5880 |
| ACF +Soil | 5603 | 4701 | 6008 | 6801 | 5395 | 6769 | 4882 | 6196 | 4877 | 5998 |
| PCF −Soil | 0.87 | 0.58 | 0.88 | 0.80 | 0.79 | 0.80 | 0.70 | 0.80 | 0.69 | 0.60 |
| PCF +Soil | 0.86 | 0.62 | 0.88 | 0.87 | 0.77 | 0.84 | 0.67 | 0.82 | 0.66 | 0.61 |

Note: ACF and PCF refer to area carbon footprint and product carbon footprint, and −Soil and +Soil refer to excluding and including soil emission and sequestration, respectively.

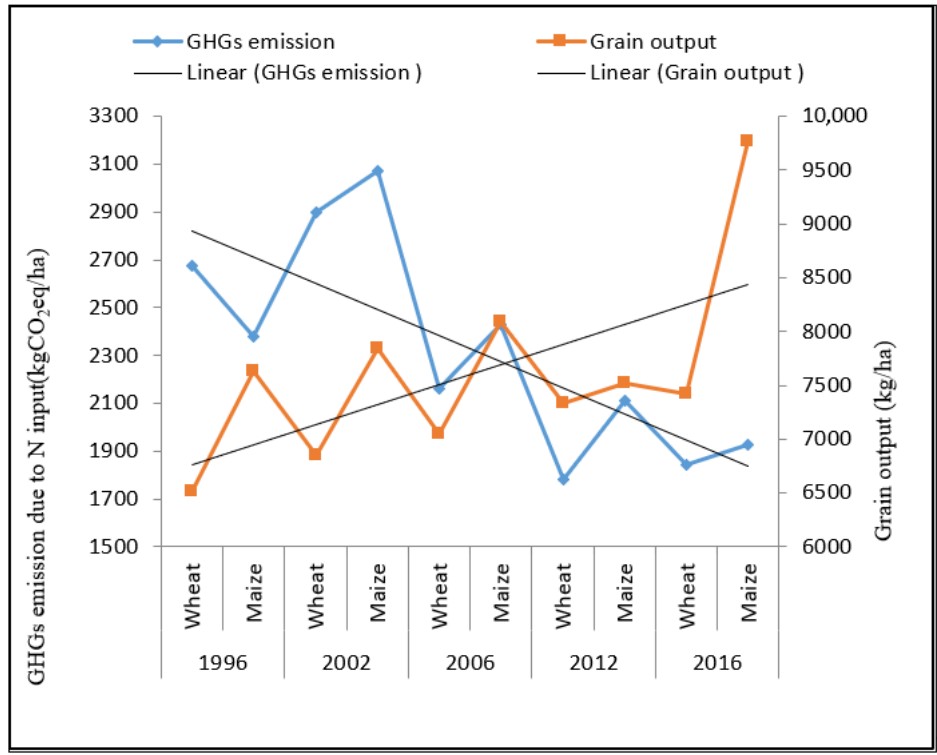

**Figure 2.** GHG emissions (kg $CO_2$eq/ha) due to N input and yield (kg/ha) of WWSM rotation system in Huantai county from 1996 to 2016.

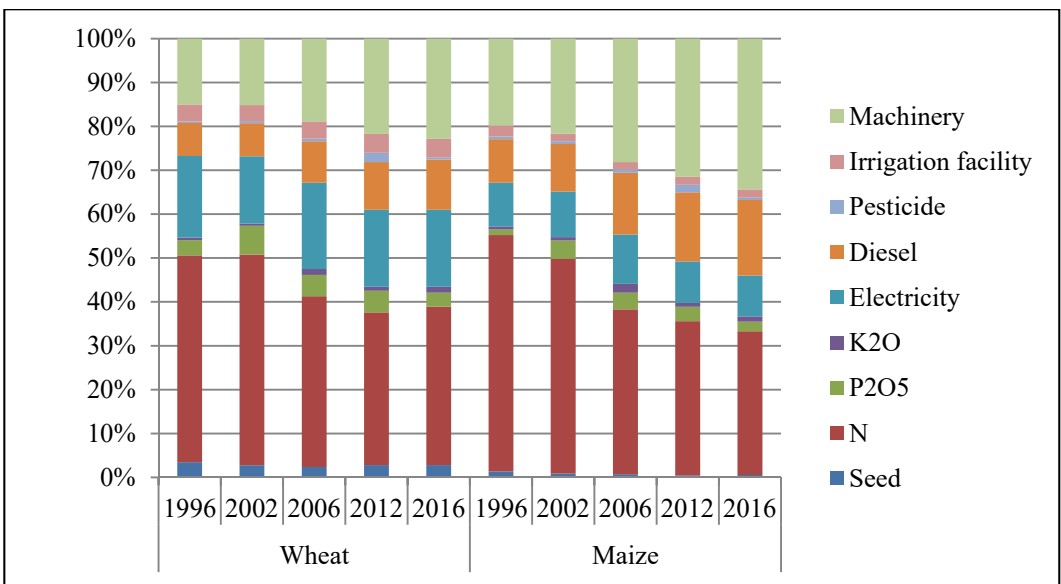

**Figure 3.** The contribution of material inputs to the carbon footprint of the WWSM rotation system in Huantai county from 1996 to 2016.

As for grain output, the yield of wheat production increased by 14%, from 6510 kg/ha in 1996 to 7425 kg/ha in 2016, and the corresponding yield increase for maize was 28%, from 7630 to 9767 kg/ha, namely 0.7% and 1.4% increase each year for wheat and maize production. Except for some atypical years (such as maize production in 2012), the trend for WWSM rotation system has been for increasing yields over the last 20 years (Table 3, Figure 2).

*3.2. Carbon Footprint Analysis*

Including soil emission and sequestration, the area carbon footprints (ACF +Soil) of wheat production was 5603 kg $CO_2$eq/ha in 1996, rising to 6008 kg $CO_2$eq/ha in 2002, and then gradually falling to 5395, 4882, and 4877 kg $CO_2$eq/ha in 2006, 2012, and 2016, respectively (Table 3 and Figure 4a). If soil emissions and sequestrations were excluded (ACF −Soil), the values of ACF were very similar. This is due to soil sequestration largely offsetting soil emissions (Table 3). For wheat, the product carbon footprints (PCFs) including soil factors (PCF +Soil), ranged from 0.66 to 0.88 kg $CO_2$eq per kg grain. If soil factors were excluded, the values ranged from 0.69 to 0.88 kg $CO_2$eq per kg grain. The trends were similar for PCFs as for ACFs, with a peak in 2002 followed by consistent decreases thereafter.

For maize production, the ACF including soil emissions and sequestrations (ACF +Soil) was 4701 kg $CO_2$eq/ha in 1996. A peak occurred in 2002, after which the ACF gradually declined to values of 6769, 6196, and 5998 kg $CO_2$eq/ha in 2006, 2012, and 2016, respectively (Table 3, Figure 4b). The value in 2016 remained above the value in 1996. If soil factors were excluded (ACF −Soil), a similar trend was observed. In contrast to wheat production, the values of ACF −Soil for maize were lower than that of ACF +Soil, due to the value of soil emission being higher than soil sequestration. However, the gap was becoming smaller over time. For the PCF of maize, the values of PCF +Soil were 0.63, 0.87, 0.84, 0.82, and 0.61 kg $CO_2$eq/kg grain over the period 1996 to 2016 (Figure 4b).

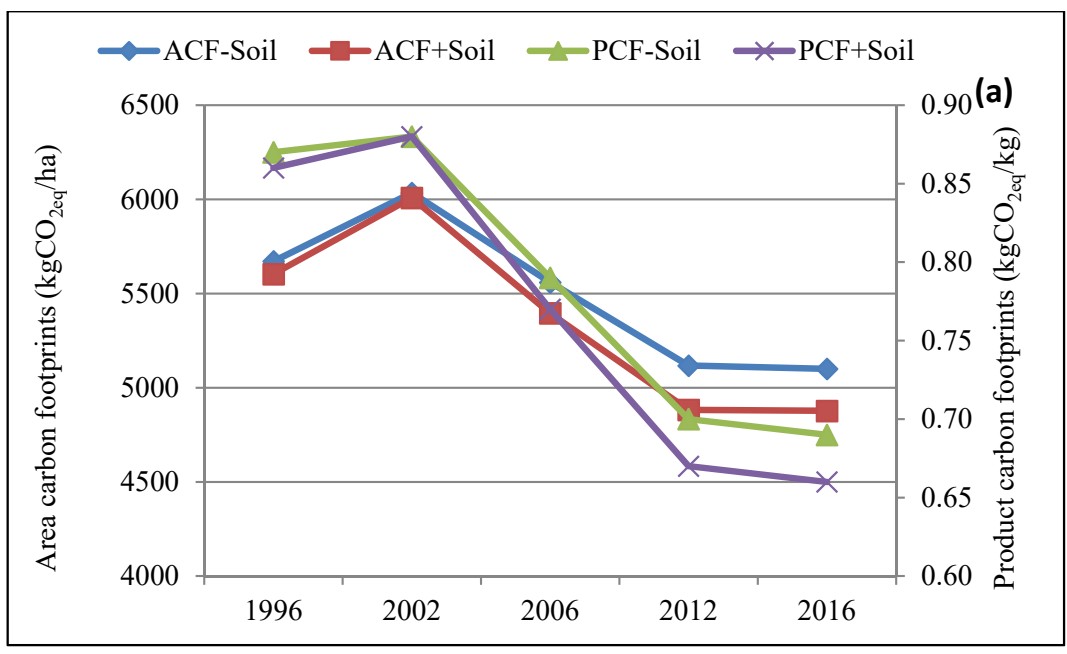

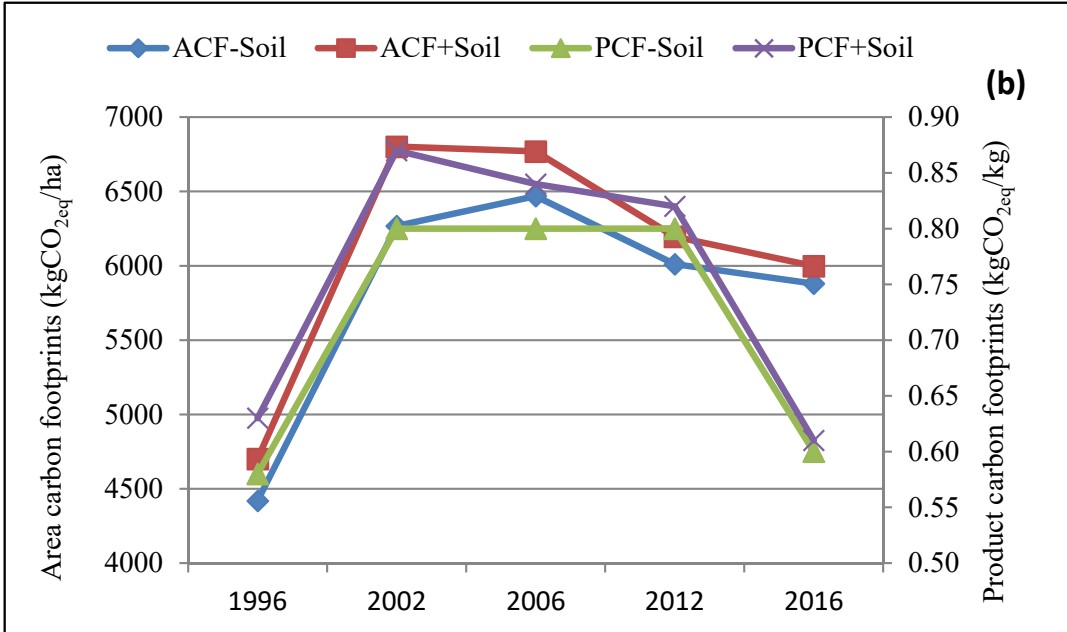

**Figure 4.** Area carbon footprint (ACF) and product carbon footprint (PCF) of wheat (**a**) and maize (**b**) rotation system in Huantai county from 1996 to 2016. Note: −Soil and +Soil refer to excluding and including soil emission and sequestration, respectively.

### 3.3. Carbon Efficiency Analysis

In the process of wheat–maize production, the CE indicators (product, ecological, and economic efficiency) initially decreased and then steadily increased over the last two decades (Table 4, Figure 5). For example, product efficiency (Ep, kg grain/kg $CO_2$eq) of wheat production, decreased from 1.16 in 1996 to 1.14 kg in 2002 and then increased to 1.31, 1.50, and 1.51 in 2006, 2012, and 2016, respectively. For maize production, the corresponding values were 1.62, 1.15, 1.19, 1.21, and 1.63 kg grain/kg $CO_2$eq, respectively. Taking 2002 as a baseline, by 2016, Ep rose by 32.5% and 41.7% for wheat and maize, respectively.

**Table 4.** Production efficiency (Ep), ecological efficiency (Ec), and economic efficiency (Ee) of winter wheat–summer maize rotation system of Huantai county, Shandong Province, China (1996 to 2016).

| | 1996 | | 2002 | | 2006 | | 2012 | | 2016 | |
|---|---|---|---|---|---|---|---|---|---|---|
| | Wheat | Maize | Wheat | Maize | Wheat | Maize | Wheat | Maize | Wheat | Maize |
| Ep (kg grain/$CO_2$eq) | 1.16 | 1.62 | 1.14 | 1.15 | 1.31 | 1.19 | 1.50 | 1.21 | 1.51 | 1.63 |
| Ec (kg $CO_2$/$CO_2$eq) | 4.63 | 6.78 | 4.55 | 4.82 | 5.21 | 4.99 | 5.99 | 5.07 | 6.07 | 6.80 |
| Ee (Yuan/$CO_2$eq) | 1.88 | 1.85 | 1.16 | 1.06 | 1.88 | 1.51 | 3.42 | 2.49 | 3.35 | 3.09 |

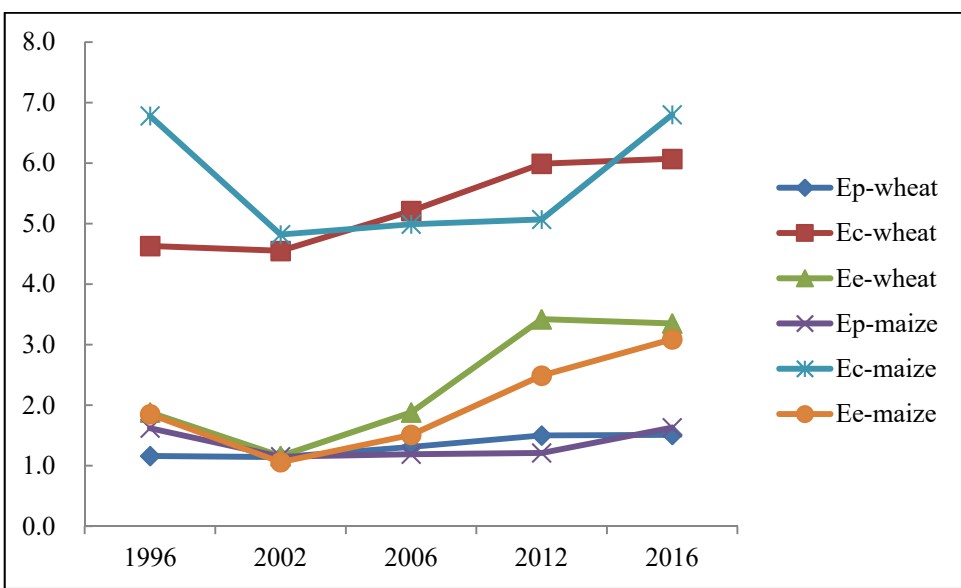

**Figure 5.** Product (Ep), ecological (Ec), and economic efficiency (Ee) of wheat–maize rotation system in Huantai county from 1996 to 2016.

The ecological efficiency (Ec) of wheat and maize production performed better than product efficiency, and the values were in the range of 4.55–6.07 and 4.82–6.80 kg $CO_2$/kg $CO_2$eq, respectively. This means C output was higher than C input, and for most of the production period the efficiency of maize was higher than that of wheat due to its higher yield of grain and biomass. In 2016, the improvement from 2002 was 33% for wheat and 41% for maize. For economic efficiency (Ee), the corresponding improvement from 2002 to 2016 was 189% and 192% (Table 4, Figure 5).

The product and ecological efficiencies were lowest in 2002 and the most important reason was the N fertilizer input, which reached its peak at this time (Figure 2). Over the following period (2002–2016), N inputs gradually decreased while yield simultaneously slowly increased (Figure 2). Thus, the two indicators were gradually improved. However, the effect of decreased N inputs was offset to an extent by an increase in machine use and diesel consumption over time (Figure 5). As for economic efficiency, not only did the N input peak in 2002, the sale price of grain was also in a trough [61]. Therefore, this indicator was also at the lowest point in 2002. Over the period of 2002–2016, not only did N inputs decrease and yield increase, another important factor was an improving grain price, that resulted, at least in part, from government provided agricultural subsidies to maintain the stabilization of the grain market price [59]. Thus, the magnitude of increase of economic efficiency was higher than that of product and ecological efficiency (Figure 5).

## 4. Discussion

### 4.1. Comparison and Analysis of Carbon Footprint

Carbon footprints of agricultural ecosystems have been widely studied over the past two decades (Table 5). What emerges from the literature is that although product carbon footprints have generally been trending downward, this has often been achieved through agricultural intensifications that have led to higher area-based GHG emissions. For example, Cheng et al. [13] used national statistical data in China for the period 1993–2007 to estimate the CFs of crop production, and the results showed that for the periods 1993–1997, 1998–2002, and 2003–2007, the carbon emissions per ha cultivated were 2530, 2824, and 3154 kg $CO_2$eq, whereas the CFs per kg product were 0.47, 0.40, and 0.39 kg $CO_2$eq. The former was increasing while the latter was decreasing over this period. Huang et al. [50] quantified the carbon footprints of rice, wheat, and maize production in China over the period of 1978–2012. The results showed that area-scaled CFs of the three crops' production systems increased from 1286, 937, and 895 kg $CO_2$eq/ha in 1987 to 2682, 2978, and 2294 kg $CO_2$eq/ha in 2012, respectively. Meanwhile, the yield-scaled CFs of rice, wheat, and maize decreased. Similar findings have been reported by Wang et al. [48] and by Xu and Lan [62].

From a global perspective, Bennetzen et al. [52] found that since 1970 the PCF of crops had decreased by 39%. Further, they forecast an addition 25% decrease in PCF to 2050. However, despite these impressive improvements in the GHG emission intensity of crop products, the GHG emissions per unit cropland has risen by 15% since 1970 and will likely increase a further 7% by 2050. What this means is that aggregated GHG emissions from cropland are increasing, which is not consistent with global efforts to stabilize the climate. As such, the evidence emerging in this study from Huantai county is important because it demonstrates that concurrent reductions in PCF and ACF are possible. In Huantai county, the total cultivated area is constrained by demand for urban and industrial land. Therefore, with agricultural inputs having reached a threshold [10,58], there now exists the possibility of reductions in total emissions from the cropping sector.

**Table 5.** Comparison of product carbon footprint (PCF) and area carbon footprint (ACF) results.

| Study Area | System Boundary | PCF (kg CO$_2$eq/kg) | | ACF (kg CO$_2$eq/ha) | | Major Source of CF | References |
|---|---|---|---|---|---|---|---|
| | | Wheat | Maize | Wheat | Maize | | |
| China | Cradle to gate +SE +SQ | 0.66 | 0.62 | 4902 | 6022 | | This study (year in 2016) |
| China | Cradle to gate +SQ | 0.67 | 0.62 | 3707 | 4436 | | [4] |
| China | Cradle to gate +SE +SQ | 0.5 | 0.4 | 2800 | 2707 | FM (42–44%), N$_2$O (32–37%) | [62] |
| China | Cradle to gate +SE −SQ | 0.30–0.46 | 0.26–0.37 | | | FMU (~90%) | [33] |
| China | Cradle to gate +SE −SQ | 0.51 | 0.44 | 2914 | 2866 | NM + N$_2$O (78%) | [13] |
| Eastern China | Cradle to gate −SQ | 0.66 | 0.33 | 3000 | 2300 | NMU (75–79%), DU (14–15%) | [35] |
| China | Cradle to gate +SE +SQ | 0.45 | 0.32 | | | FM (65%), N$_2$O (26%), DU (9%) | [62] |
| North China | Cradle to gate +SQ (−SQ) | 0.23–0.24 (−SOC) −0.02–0.3 (+SOC) | 0.43 (−SOC); 0.13–0.37 (+SOC) | | | FMU (45–49%), EC (35–43%) | [38] |
| China | Cradle to gate −SQ | 0.35–0.62 | 0.2–0.4 | 940–2980 | 900–2290 | FMU (68–76%), EC (17–23%) | [50] |
| North China | Cradle to gate +SQ | 0.32 | 0.45 | | | NMU (49%), EC (45%) | [32] |
| China | Cradle to gate −SQ | 0.27 | 0.23 | | | | [48] |
| America | Cradle to gate +SE −SQ | 0.28 | | | | FM (24%), N$_2$O (50%), DU (19%) | [21] |
| UK | Cradle to gate −SE −SQ | | | 2807 | | NMU (83%) | [26] |
| Australia | Cradle to port +SE −SQ | 0.3 | | | | FM (30%), N$_2$O (9%), T (12%) | [17] |
| Australia | Cradle to port +SE −SQ | 0.4 | | | | N$_2$O (60%) | [18] |
| New Zealand | Cradle to gate −SE −SQ | 0.1 CO$_2$ | | 1032 CO$_2$ | | FM (52%), DU (20%) | [20] |
| Sweden | Cradle to gate −SQ | 0.38 | | | | | [30] |
| EU and USA | Cradle to port −SQ | 0.58 | 0.67 | | | | [30] |
| UK | Cradle to gate −SQ | 0.8 | | | | NMU (70%) | [28] |
| Sweden | Cradle to gate +SE +SQ | 0.31 | | | | FM (21%) and N$_2$O (70%) | [23] |
| France | Cradle to gate −SQ | 0.45, | 0.4 | | | | [25] |
| UK and Spain | | 0.52 (UK), 0.58 (SP) | | | | | [24] |
| North Iran | Cradle to gate +SE −SQ | 0.33 | 0.17 | 1171 | 1441 | DU (25–46%), N$_2$O (15–38%), EU (40%) | [19] |
| Denmark | Cradle to gate +SE +SQ | 0.39 | | | | FM (35%), N$_2$O (47%), DU (19%) | [22] |
| South Africa | Cropland emission −SQ | 0.11 | 0.14 | | | | [27] |
| Canady | Cradle to gate +SE −SQ | 0.38 | 0.33 | | | FMU (81%) | [63] |
| Italy | Cradle to gate +SE −SQ | 0.45 | 0.45 | | | FMU (66–73%) | [64] |
| Slovenia | Cradle to gate −SQ | 0.11–0.15 | 0.23–0.25 | | | FM (42–76%) | [65] |
| Globe | | 0.58 | 0.49 | 2165 | 2954 | | [31] |
| Globe | | 0.52 | 0.47 | | | | [29] |

Note: EU and SP refer to the European Union and Spain. SE and SQ refer to soil emission and soil organic carbon sequestration. FM, FMU, NMU, DU, EC, T, and N$_2$O refer to fertilizer manufacture, fertilizer production and use, N fertilizer manufacture and use, diesel use, electricity consumption, transportation, and cropland N$_2$O emission.

*4.2. Comparison and Analysis of Carbon Efficiency*

In this study, product, biomass, and economic efficiency are used to evaluate the carbon efficiency of cropping systems over time. Economic efficiency was found to have high volatility highly due to the influence of market price. However, the results showed that, since 2002, all the three indicators have been steadily increasing. There exists a variety of comparable studies both in China and other regions [5,8,38,41,42,66,67], and the results obtained for Huantai county are within the range of values reported elsewhere in China; but in America, Canada, and Europe, the efficiencies have been 2–5 times higher than that in China including Huantai and other regions (Table 6). Certainly, it is difficult to directly compare results across different studies due to differences in system boundary definition and inconsistencies in emission coefficients used. That said, compared to other regions, the product efficiency of cropping in China appears lower, suggesting there exists greater potential for efficiency increases.

**Table 6.** Carbon efficiency of cropping production in China and other regions.

| Study Area | Cropping System | Product Efficiency (kg grain/kg $CO_2$eq) | Ecological Efficiency (kg biomass/kg $CO_2$eq) | Economic Efficiency (Yuan/kg $CO_2$eq) | Reference |
|---|---|---|---|---|---|
| China | Wheat | 1.51 | 6.07 | 3.35 | This study (year in 2016) |
| | Maize | 1.63 | 6.80 | 3.09 | |
| China | Wheat | 2 | 8.94 | 3.52 | [41] |
| | Maize | 4.06 | 13.68 | 5.48 | |
| China | Wheat–Maize Rotation | 0.53 | 3.11 | 1.67 | [44] |
| China | Wheat–Maize Rotation | 0.29 | 0.94 | 0.6 | [42] |
| China | Crop product | 1.95–2.48 | | | [8] |
| China | Wheat | 0.99 | | 2.56 | [51] |
| | Maize | 1.26 | | 2.94 | |
| China | Wheat | 1.52 | | | [35] |
| | Maize | 3.03 | | | |
| China | Wheat | 1.39–1.53 | 7.6–8.6 | | [38] |
| | Maize | 4.13–4.39 | 19.3–19.7 | | |
| China | Wheat | 1.96–2.5 | | | [5] |
| | Maize | 2.7–3.1 | | | |
| America | Wheat | 2.86–4 | | | [66] |
| | Maize | 4–8.3 | | | |
| Canada | Wheat | 3.7–4 | | | [67] |
| India | Wheat | 8.3 | | | [68] |
| Slovenia | Wheat | 6.7–9.1 | | | [65] |
| | Maize | 4.3–4.5 | | | |

*4.3. Key Factor Analysis*

In Huantai county, the WWSM rotation system has been characterized by increasing yields over time. There have also been changes in material inputs, namely decreases in N fertilizer inputs and increases in machinery use and diesel consumption. These changes in farming inputs can be linked to two major local government interventions, one being the promotion of straw incorporation, the other related to soil testing and precision fertilizer management. Since 1980, farmers in Huantai county have gradually adopted the practice of incorporating crop residues into farmland. By 2008, it is estimated that 90% of straw including wheat and maize was incorporated into cropland in this region. This appeared about 10 years earlier than in other regions of China [57,59]. Liao et al. [53,54] made a series of studies about cropping systems in Huantai county and identified that over the past 30 years (from 1982 to 2011), soil organic carbon content of topsoil (0–20 cm) increased by 41%, and its density by 57% as well. Additionally, since 2011, fertilizer use

has been managed more closely as soil testing was introduced and has become widely accepted [8,57,69]. Zhang et al. [57] found that from 1980 to 2014, the value of reactive N losses decreased 21.5%, and at the same time, the annual N recovery increased from 39.8% to 54.1%. Zhang et al. [57] and Liang et al. [59] made an N balance analysis from 1996 to 2012, and the results showed that indirect N inputs, especially straw return, played an important role to keep the N balance and made some contributions to the increases of soil organic carbon.

N application is a critical factor for increasing the yield of cropping systems. Tilman et al. [2] forecasted that, in order to double crop production, the N fertilizer consumption will need to increase by 140%, namely from 104 Mt/year globally in 2010 to 250 Mt/year in 2050. Mueller et al. [6] identified that only a 9% increase in N fertilizer consumption would make a 30% increase in production of the major cereals. Cui et al. [3] reviewed a series of experiments in China and indicated that with a 38% increase in N fertilizer application, the yield and GHG emissions per area will increase by 70% and 37%, while the GHG intensity per unit product would decrease 19%. In contrast, modeling undertaken by Chen et al. [4,8] demonstrated the potential to increase crop yields by 30% to 50% in China without additional N inputs, challenging the dominant paradigm concerning crop yields and N inputs. What is important about the multi-decadal case study evidence from Huantai county is that it validates the earlier modeling of Chen et al. [8]. Reducing N fertilizer applications while simultaneously increasing indirect N inputs through straw return and biological fixation is an effective approach to improve crop yields in farmland ecosystems in the future.

### 4.4. Limitation and Uncertainly Analysis

This study, based on the data from longitudinal farm surveys and local farming system experiments, used the life cycle assessment method to make a detailed analysis of the CF and CE of cereal production for Huantai county. The system boundary was from cradle to farm gate, including all of the major agricultural inputs, on-farm activities, and soil processes relating to GHG emissions and carbon sequestration. It has already been mentioned that comparisons between CF and CE studies are difficult due to the myriad of modeling choices that are possible. We note that the GHG emission factor used in this study for N fertilizer production is higher than what has been reported in other studies [5,41,62]. However, this is unlikely to materially impact study conclusions. There are large uncertainties associated with the assessment of changes in SOC. However, we used the best available data from long-term field trials located in the region. That said, there is uncertainty about whether SOC can be increased into the future at the same rate as has been recorded in the past.

### 5. Conclusions

Agricultural production is central to global food security, and an increasing world population creates a requirement for increasing crop yields. However, climate stabilization goals necessitate absolute reductions in agricultural GHG emissions. Most of the evidence concerning cereal production suggests that important reductions in product carbon footprints have been realized. However, these have been realized through agricultural intensifications that have increased area-based GHG emissions, thereby not contributing to the goal of reducing absolute emissions. This long-term analysis of the WWSM rotation system in Huantai county, based on longitudinal farm surveys and local farming system experiments, has demonstrated the potential of reducing both product-based and area-based GHG emissions. The critical factors were reduced N fertilizer use informed by soil nutrient testing, along with high levels of residue incorporation. As Huantai country is broadly representative of high-yielding, intensive cropping across the North China Plain, it would seem possible to extend such farming practices on a large scale.

**Author Contributions:** L.L. and B.G.R. designed and wrote the paper, and L.W. provided some pictures and analyzed the data. All authors have read and agreed to the published version of the manuscript.

**Funding:** This research was funded by the Theoretical Innovation Program of Joint Association of Social Science in Guizhou Province, grant number GZLCZB-2019-005.

**Institutional Review Board Statement:** Not applicable.

**Informed Consent Statement:** Informed consent was obtained from all subjects involved in the study.

**Data Availability Statement:** Data available in a publicly accessible repository.

**Acknowledgments:** The authors would like to thank the anonymous reviewers for their precious comments.

**Conflicts of Interest:** The authors declare no conflict of interest.

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
