# Peer review of "Food Security and Climate Stabilization: Can Cereal Production Systems Address Both?"

_sustainability, doi:10.3390/su13031223_

Round 1

Reviewer 1 Report

This is an interesting article and it gives hope to world food production and mitigation of climate change despite the obvious need to increase the production of food.

I have some comments to you:

The letter indices in the author names are not explained in the addresses, please add them.

There is a big collection of references in the Introduction – less would suffice.

There are some mistakes in the references in the text. They need to be carefully checked.

Language is mostly good, but sudden lapses occurs. I have corrected some of them, but one more check/editing is needed.

Reference list needs editing, the style varies and there are also mistakes. I have attracted attention to some of them.

The results (lines 292-337) seem to mix N input and its CO2-equivalent input, which is confusing.

Results on lines 372-376 are written vice versa than the real results and have to be corrected.

Further comments are in the pdf attached.

Author Response

Q1:The letter indices in the author names are not explained in the addresses, please add them.

A1:We have revised this problem, and thanks for you remind.

Q2: There is a big collection of references in the Introduction – less would suffice.

A2:We have reduced some references according to your advice.

Q3:There are some mistakes in the references in the text. They need to be carefully checked.

A3: We have checked all references in the text, and revised some references in the text, and we hope this manuscript would reach the standard of this journal.

Q4:Language is mostly good, but sudden lapses occurs. I have corrected some of them, but one more check/editing is needed.

A4:I and my co-author have carefully checked our manuscript and we hope the revised version would meet the requirement of our journal.

Q5:Reference list needs editing, the style varies and there are also mistakes. I have attracted attention to some of them.

A5: We have revised all references and added some references according to the text.

Q6:The results (lines 292-337) seem to mix N input and its CO2-equivalent input, which is confusing.

 A6:This is a serious mistake, and thanks you remind and we have revised this problem including sentence and figure.

Q7:Results on lines 372-376 are written vice versa than the real results and have to be corrected.

 A7:This is a mistake and we have corrected it and thanks for your remind.

Further comments are in the pdf attached.

The following are some explain about you questions in the pdf attachment.

A8:Line 108 and 112, reviewer said he did not understand this sentence and we discussed the problem. In fact, in the two sentences, we want to express that it is impossible to realize the climate stabilization if only depending on GHGs emission reduction per unit agricultural product. But if the emission reduction per unit area have been realized, the agriculture would make a real contribution. Therefore, the conventional model should be changed, namely the agricultural inputs should be reduced while the yield should increase at present and in future.

A9: In Fig 1, we have changed this original picture, and in new picture the three towns investigated have been labeled.

A10:Line 176, as for the inputs change so much in Table 1, before 2002, most of farmers trusted the idea that getting more output must depend on more input, and after this year, the government promoted two policies in this county, one is soil test and precision fertilization, and the other is crop residues into farmland. Therefore, the inputs of fertilizer have been decreasing while the diesel consumption was drastically been increasing due to smash straw. As for electricity use, the water use of winter wheat-summer maize(WWSM) rotation system  depend on underground water. Due to the differences of precipitation among different years, and then, the electricity consumptions were different, and in North China Plain, there precipitation is unstable, and rainfall become less and less over the last two decades, therefore, the WWSM need more electricity which is a problem need to be solved in future.

A11:In Table 3, the reviewer asked why the CH4 sequestration and soil sequestration are the same. In section 2.4.2, based on some experimental results from 1981 to 2011,Zhao (2017), Zhao et al (2017) and Liao et al (2015) according to Eq (4-6)calculated the CH4 absorption by agricultural soils and soil sequestration and did not consider biomasses. Therefore, we took an averaged value for every season and every year.

Reviewer 2 Report

The article is valuable, congratulations to the autors.

Below, please find my comments/remarks.

141 - The yield in 1990 was >15Mg/ha of grain across the entire region. What is this yield? Corn’s yield would be very high and wheat unavailable.

301 - the N fertilizer input increased from 2674 kg/ha in 1996 to 2898 302 kg/ha in 2002; should be 267.4 kg/ha in 1996 to 289.8 kg/ha in 2002

Fig. 2 the values on Y axis are too high

303 - amount was 1843 kg/ha; should be 184.3 kg/ha

335 - 558kgCO2eq/ha; should be 558 kgCO2eq/ha

Author Response

The article is valuable, congratulations to the authors.

Below, please find my comments/remarks.

Q1:141 - The yield in 1990 was >15Mg/ha of grain across the entire region. What is this yield? Corn’s yield would be very high and wheat unavailable.

A1:In Huantai county, the major production model is winter wheat-summer maize rotation system, and the total yield of wheat and maize grain is >15Mg/ha, the yield of one cropping system could not >15Mg/ha.

Q2:301 - the N fertilizer input increased from 2674 kg/ha in 1996 to 2898 kg/ha in 2002; should be 267.4 kg/ha in 1996 to 289.8 kg/ha in 2002

A2:This is a mistake, we mixed N input and its CO2-equivalent input, and we have corrected it and thank you remind.

Q3:Fig. 2 the values on Y axis are too high

 A3:In Fig.2 the values on Y axis were designed according these data including GHGs emission and total yield including wheat and maize product.

Q4:303 - amount was 1843 kg/ha; should be 184.3 kg/ha

 A4:As mentioned above, we mixed N input and its CO2-equivalent input, and we have corrected it.

Q5:335 - 558kgCO2eq/ha; should be 558 kgCO2eq/ha

 A5:I think this would be no problem.